# Relationships between topographic factors, soil and plant communities in a dry Afromontane forest patches of Northwestern Ethiopia

Liyew Birhanu[1]*, Tamrat Bekele[2], Binyam Tesfaw[3], Sebsebe Demissew[2]

1 Department of Biology, College of Natural and Computational Sciences, Debre Markos University, Debre Markos, Ethiopia, 2 Department of Plant Biology and Biodiversity Management, Addis Ababa University, Addis Ababa, Ethiopia, 3 School of Earth Science, Addis Ababa University, Addis Ababa, Ethiopia

* liyewmtu@gmail.com

**Data Availability Statement:** All data are available with in the paper.

**Funding:** This research was funded by Addis Ababa University. The funders had no role in the

## Abstract

Plant community types are influenced by topographic factors, the physical and chemical properties of soil. Therefore, the study was carried out to investigate the relationships of soil and topographic factors on the distribution of species and plant community formation of the Dega Damot district in Northwestern Ethiopia. Vegetation and environmental data were collected from 86 plots (900 m$^2$). Agglomerative hierarchical cluster analysis and redundancy analysis (RDA) with R software were used to identify plant communities and analyze the relationship between plant community types and environmental variables. Five plant community types were identified: *Erica arborea-Osyris quadripartita*, *Discopodium penninervium-Echinops pappii*, *Olea europaea -Scolopia theifolia*, *Euphorbia abyssinica-Prunus africana*, *Dodonaea anguistifolia-Acokanthera schimperi*. The RDA result showed that the variation of species distribution and plant community formation were significantly related to altitude, organic matter, moisture content, slope, sand, pH, EC, total nitrogen and phosphorus. Our results suggest that the variation of plant communities (Community 1, 2, 3, and 4) were closely related to environmental factors, including altitude, moisture content, OM, slope, sand, pH, EC, soil nitrogen, and phosphorus, among which altitude was the most important one. However, all the measured environmental variables are not correlated to *Dodonaea anguistifolia-Acokanthera schimperi* community type. Therefore, it can be concluded that some other environmental variables may influence the species composition, which is needed to be further investigated.

## Introduction

The existence of plant communities is due to the interaction between plant species and their environment [1]. Plant community distribution pattern is influenced by many environmental factors such as climate, soil nutrients and topographic features [2–5]. Consequently, the relationship between the distribution of plant communities and environmental factors is one of the most important research problems in plant ecology [6, 7].

study design, data collection and analysis, decision to publish or preparation of the manuscript.

**Competing interests:** The authors have declared that no competing interests exist.

At regional and global scales, plant species responses are related to climatic factors [8, 9], while at local scales, topographic and edaphic factors play critical roles in controlling plant community formation [10–12]. According to [13] and [14], soil and topography are known as important factors affecting vegetation distribution. Therefore, environmental variables are important not only in identifying plant community structure and species distribution variations at a spatial scale but also in providing insight into the environmental requirements of the plant species needed for successful ecological restoration and biodiversity protection [15, 16].

In Ethiopia, different plant ecological studies have been conducted on the distribution of plant community types associated with environmental factors [e.g. 17–23]. However, no study is known about the relationships between environmental factors and plant species distribution and community formation in the Dega Damot district of Northwestern Ethiopia. This lack of knowledge hampers targeted conservation and restoration efforts [24]. Therefore, the objective of this study was to study the classification of plant communities and assess the relationship between plant community types and environmental variables in the study area.

## Materials and methods

### Description of the study area

The study was carried out in a dry Afromontane forest landscape in Dega Damot district, Amhara Regional State, in the Northwestern part of Ethiopia. The major nearby town is Feres-bet which is 399 km away Northwestern of Addis Ababa. The district is bordered by Dembe-cha (in the south), JabiTehnan (in the southwest), Quarite (in the west), HuletIju Enesie (in the north) and Bibugn (in the east) districts. The study area extends between 10°44′41.69″N and 11°00′10.56″N, and 37°26′58.43″E and 37°42′35.42″E (Fig 1). Degadamot district covers approximately 83,124 hectares of land area [25]. The district has forest patches. Of them, five forest patches were selected for the study which are relatively undisturbed in the district. Topographically the study area consists of mountainous areas, undulating, valleys, and plains [25,

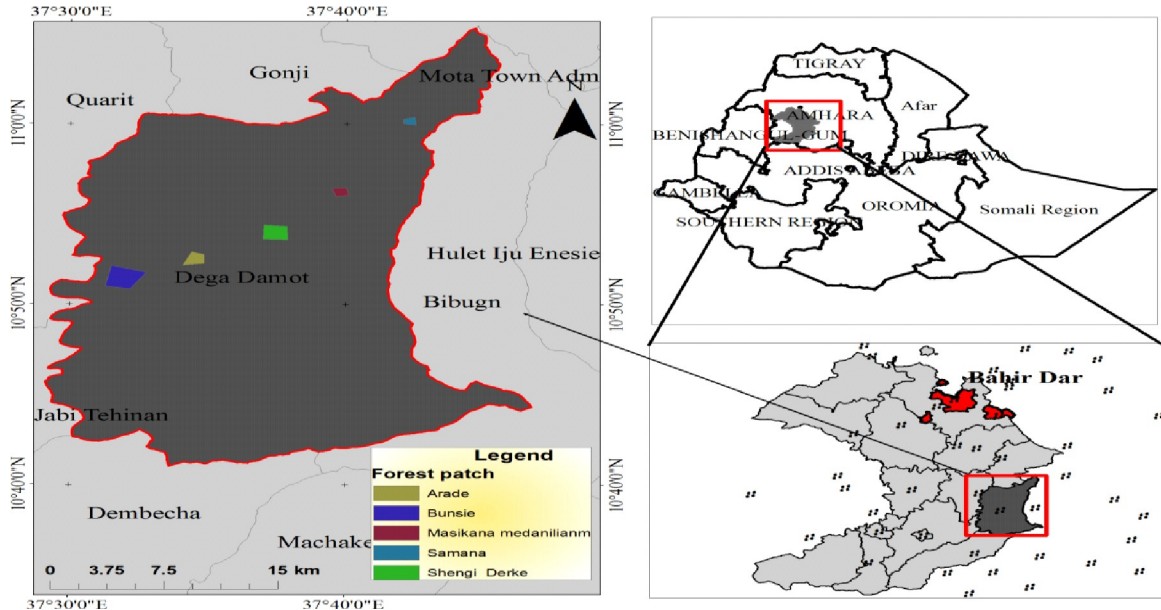

**Fig 1. Location of forest patches in Dega Damot district, the star shows the small towns in the zone and in the district.** Our study does not need to supply a copy right notice for Fig 1. Because the location map of the study area (Fig 1) shape file data was obtained from Ethiopian Mapping Agency (https://africaopendata.org/dataset/ethiopia-shapefiles) is free and open to researchers.

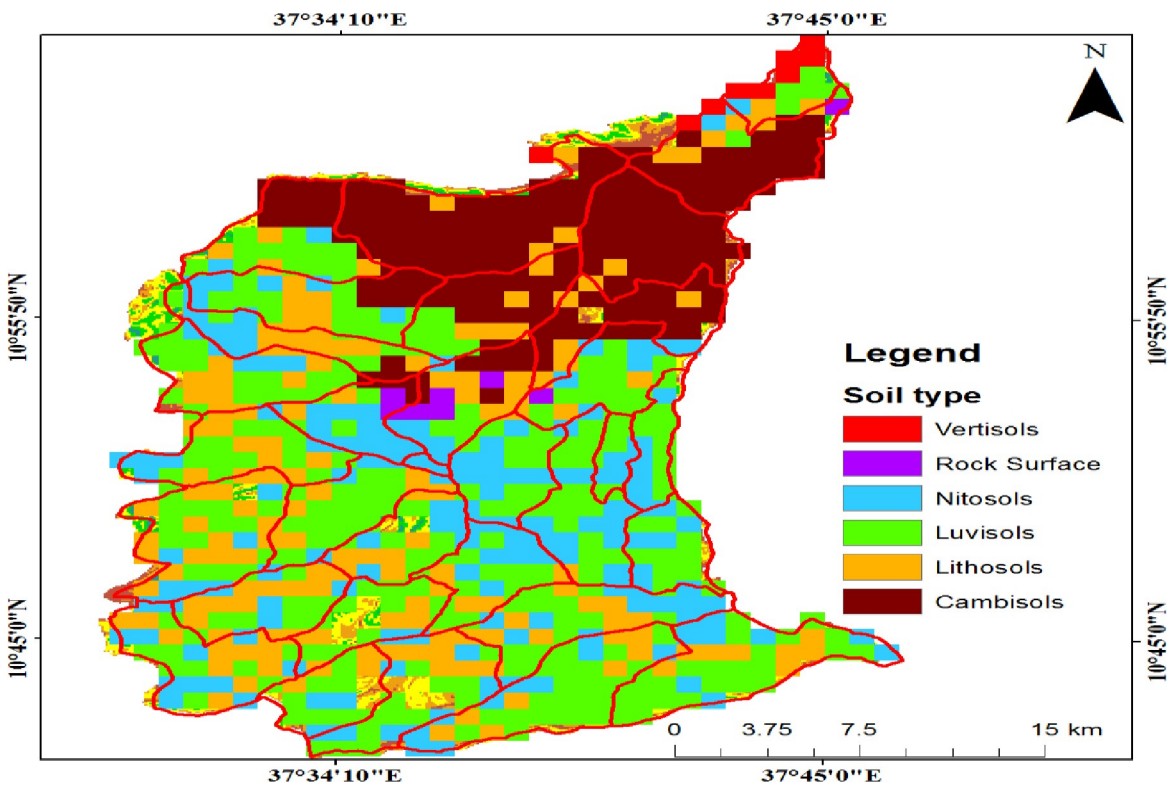

**Fig 2. Soil map of Dega Damot district (data source: MoWIE), our study does not need to supply a copy right notice for Fig 2.** Because the shape file data for soil map obtained from the Geographic Information System (GIS) department of the Ministry of Water, Irrigation and Electricity (MoWIE), is free and open to researchers.

26]. The elevation of the study area ranges from 1738 to 3586 m above sea level. On the other hand, the slope classes range from 0–50 degrees [25].

The average monthly maximum temperatures were 24.7 $^0$C and monthly average minimum temperatures 9.2 $^0$C. Annual rainfall and temperatures were 2113 mm and 15.9 $^0$C, respectively [25]. According to the data obtained from the Geographic Information System (GIS) department of the Ministry of Water, Irrigation and Electricity (MoWIE; [27], the dominant soil type of the study area are Luvisols (28523.31ha) and Cambisols (17552.81 ha) (See Fig 2 and Table 1). The vegetation of the study area belongs to the categories of dry evergreen Afromontane vegetation [28]. The area is covered by agricultural lands, settlements, forestland, shrubland bare lands and grazing lands. Agriculture is common practice in the Dega Damot district, with typical crops of *Hordeum vulgare* (barley), *Triticum sp.*(wheat), *Vicia faba* (faba bean), *Eragrostis tef* (teff), *Zea mays* (maize) and *Solanum tuberosum* (potato) [25, 26]. We obtained written permission from Agriculture and Rural Development Office of the Dega Damot district to collect both vegetation and environmental data and conduct our research.

## Reconnaissance survey and sampling technique

A reconnaissance survey was made in January 2017 across the forest patches to get an impression of the site conditions and identify the sampling sites in the study area. The forest patches were located at different altitudes, ranging from 1881–2947 m and also occupy different sizes ranging from 50 ha to 405 ha (Table 2). The actual fieldwork was conducted between February and November 2017. A systematic sampling technique was used for vegetation and

**Table 1. Soil types and area coverage of the study area.**

| No | Major soil types | Area (ha) |
|----|------------------|-----------|
| 1 | Vertisols | 897.59 |
| 2 | Nitosols | 16256.29 |
| 3 | Luvisols | 28523.31 |
| 4 | Cambisols | 17552.81 |
| 5 | Rock surface | 797.85 |
| 6 | Lithosols | 16455.76 |

environmental data collection. Sampling sites were arranged along transects from the forest patches. The first transect and plot were selected purposely at one side of the forest by avoiding the forest edge.

A total of 22 line transects (6 each in Shangi Derke and Bunsie forest patches, 4 in Aradie forest patch and 3 each in Masikana and Samana forest patches) were laid along elevation gradients. The number of plots per transect differs depending on the length of the transect (Table 2).

The elevation between two consecutive plots and transect were 50 and 200 ma.s.l. apart in the forests, respectively. Accordingly, a total of 86 sampling plots of each measuring 900m$^2$ (30 m × 30 m) were taken from all 5 forest patches. Of which, 29 from Shangi Derke, 32 from Bunsie, 10 from Aradie, 9 from Masikana Medihaniyalem and 6 from Samana forest patches were sampled. When a study mainly focuses on only woody species composition and diversity, research has mainly ignored studies on diversity and not included herbaceous species and rare species which are important to the understanding of a comprehensive list of plant species diversity in the study area. Therefore we have included the herbaceous species. Five smaller subplots of 2 *2 m$^2$, four at the corner, one at the center of the main plot were established for herbaceous plant species collection. In our study the overall cover of herbaceous species estimated by the average of individual cover values of species taken from the subplots (4 m$^{2)}$.

## Vegetation data collection

In each plot, all individual trees and shrubs with a diameter at breast height (DBH at 1.30 m above ground) >2.5 cm were measured for DBH classes using a measuring tape. The height of woody individual species with > 3m height was measured using Suunto Clinometers. Within each plot, the cover/abundance values were estimated for all plant species. Three-column data tables were constructed in Microsoft Excel 2007 and saved in the CSV (comma delimited) format. The first, second, and third columns represented by plots, species and abundance of species, respectively. After that, the percent cover of each species was transformed to ordinal scales and assigned to one of the nine cover-abundance classes according to the modified 1–9 Braun-Blanquet scale [29]. The scales for cover-abundance values are $1 \leq 0.1\%$, $2 = 0.1$ to $1\%$,

**Table 2. Size, number of plots, transect and altitudinal range of forest patches.**

| No | Name of forest patch | Area (Ha) | No of plots | No transect lines | Length | Altitudinal range (m) | |
|----|----------------------|-----------|-------------|-------------------|--------|-------------|---------|
| | | | | | | Minimum | Maximum |
| 1 | Shangi Derke | 225 | 29 | 6 | 2.2 km | 2475 | 2947 |
| 2 | Arade | 132 | 10 | 4 | 0.5 km | 2688 | 2912 |
| 3 | Bunsie | 405 | 32 | 6 | 2.5 km | 2517 | 2906 |
| 4 | Masikana medanilianm | 75 | 9 | 3 | 0.6 km | 2375 | 2732 |
| 5 | Samana | 50 | 6 | 3 | 0.4 km | 1881 | 2110 |

3 = 1 to 2%, 4 = 2 to 5%, 5 = 5 to 10%, 6 = 10 to 25%, 7 = 25 to 50%, 8 = 50 to 75%, and 9 >75%. Finally, the three-column data of the cover-abundance value of each species were imported to R statistical software version 3.5.2 [30] and matrified to carry out the cluster analysis and other parameters such as for ordination, diversity and evenness. The packages labdsv" and vegan" were used to transform the cover values of the vegetation data into the modified Braun Blanquet 1–9 scale and matrified the abundance data.

Moreover, species occurring outside the plots were also recorded to obtain a complete floristic list of the study area. Voucher specimens were collected, coded, pressed and dried for subsequent identification and verification at the National Herbarium (ETH), Addis Ababa University, using Volumes 1–8 of Flora of Ethiopia and Eritrea.

## Environmental data collection

Geographical data (altitude, latitude, and longitude) were recorded using GPS for each plot in the forest patches. The slope of each sample plot was measured using Suunto clinometer.

Soil samples were collected from these 86 plots for soil analysis. The samples were collected from topsoil (0–30 cm). From each plot, the soil samples were collected at five locations, one from the center and four from corners and these are mixed to produce a composite sample and only one sample was taken for analysis. The samples were air-dried, grounded, and passed through a 2-mm sieve to remove the stone pieces and large root particles before analysis. The soil analysis was assessed in a soil laboratory of Debremarkos soil research and fertility improvement center.

Soil texture, soil moisture content, bulk density, pH, EC, CEC, total nitrogen, organic matter and available phosphorus were measured for the soil data collected from the selected forest patches. The soil bulk density (BD) was measured using core method prescribed for undisturbed soils. It was was calculated as the ratio of oven-dried soil weight to the volume of the soil:

$$BD = \frac{wd}{V},$$ (1)

where wd is the weight of oven-dry soil (g) and v is the volume of the soil (cm$^3$). The volume was calculated from the volume of the core sampler. The core sample was oven-dried to a constant weight using an oven at 105˚C for 24 h. The soil texture was measured using the hydrometer method following [31]. The apparatuses used in measuring soil texture were plunger, hydrometer jar and hydrometer. Soil pH was measured by taking a 1:2.5 soil/water suspension using a glass electrode pH-meter following [32]. The electrical conductivity of the soil sample was measured by preparing a 1:2.5soil/water suspension using a conductivity-meter following [33]. Organic carbon was determined by Walkely and Black method [34]. The organic matter (OM) content was calculated by multiplying the OC content by 1.724 (OM = 1.724 x OC). Total nitrogen was ascertained by the Kjeldhal procedure after digestion with concentrated H$_2$SO$_4$ [35]. Available phosphorus (P) was analyzed by Uv/vis spectrophotometer following the Olson method [36]. Cation exchange capacity (CEC) was determined after extracting the soil samples by ammonium acetate (1N NH4OAc) at pH 7.0 [37].

## Data analysis

**Plant community classification.** Cluster analysis helps to group a set of observations (plots) based on their floristic similarities [38, 39], i.e. to determine plots that can be classified into the same groups based on the species abundance data. Therefore, for this analysis the data matrix contained 86 plots and 164 plant species were grouped using hierarchical cluster

analysis. In this study, an agglomerative hierarchical clustering (AHC) was employed using a similarity ratio (SR) and Ward method (Minimum-variance clustering) to identify plant communities. R statistical software version 3.5.2 [30] and the packages Cluster and Vegan [40, 41] were used during cluster analysis.

An indicator species analysis (ISA) was performed using the indicator value (IndVal) method in R package cluster, Vegan and Labdsv [41, 42] to identify diagnostic species for the naming of plant communities. The indicator value index (IndVal) is based on the abundance of a given species and its occurrence within a given set of samples. The statistical significance of the indicator values for each species was assessed by a Monte Carlo (randomization) test procedure. In this analysis, species with a significant indicator value of P<0.05 is considered to be an indicator species of a community. Finally, the plant community types were named after two of the dominant species that had an indicator value of p< 0.05.

**Species diversity.** Shannon-Wiener diversity indices and Shannon's evenness were computed to describe species diversity, richness and evenness of the study area, sites and plant communities [39].

$$H\prime = -\sum_{i=1}^{s} Pi \ln pi \tag{2}$$

Where H' = Shannon diversity index
s = number of species,
pi = proportion of ith species, ln = the natural logarithm.
Shannon's evenness index (J) was also calculated by using:

$$J = {}^{H'}\big/_{H}\prime max \tag{3}$$

where H' = Shannon–Wiener Diversity Index; and H' max = lns where s is the number of species in the plot.

Sørensen's similarity coefficient was used for comparing the similarity of two communities or forest patches.

$$Ss = \frac{2a}{2a + b + c} \tag{4}$$

where b and c are the no_of species in communities, forest patches or study areas b and c; a - the number of species common to both and Ss is the Sørensen's similarity coefficient.

**Ordination.** Ordination is a multivariate method that articulates the relationships between species, plotes and environmental variables in a low-dimensional space using ordination diagrams [38, 43]. A preliminary analysis of the data by Pearson's correlation coefficient was calculated to find a significant correlation between environmental variables and to remove auto-correlated variables from the ordination analysis using SAS software.

Redundancy Analysis (RDA) ordination method was used for this analysis for testing the vegetation and environmental variables relationships [41, 44]. RDA was selected due to its better visualization of the graph. In this analysis, 86 sample plots, 163 species and 10 environmental variables (altitude, slope, bulk density, moisture content, clay, pH, EC, available Phosphorus, CEC, and OM) were included. Adonis test was performed to determine the significance of the environmental variables on plant community distribution. Consequently, RDA ordination was plotted using cover-abundance values of plant species and data of significant environmental variables.

One-way ANOVA followed by post-hoc Tukey HSD test was used whether there were significant mean differences among plant communities about environmental variables, species richness, diversity and evenness.

## Results

### Floristic composition

A total of 176 species, belonging to 80 families were recorded in the study area. Asteraceae and Fabaceae were the top most dominant families. Of the total plant species composition of the forest, 18 (10.6%) species are endemic to Ethiopia and Eritrea.

### Plant community types

Five plant community types were identified from the hierarchical cluster analysis in the study area (Fig 3). The community is named after one or two dominant indicator tree or shrub species selected by the relative magnitude of their indicator values. In this study, a species is considered as an indicator of a group when its indicator value is significantly higher at $p < 0.05$ (Table 3). Consequently, the identified communities were *Erica arborea-Osyris quadripartita*, *Discopodium penninervium-Echinops pappii*, *Olea europaea* subsp. *cuspidata-Scolopia theifolia*, *Euphorbia abyssinica-Prunus africana*, *Dodonaea anguistifolia-Acokanthera schimperi* community types.

### *Erica arborea-Osyris quadripartita* community type (C1)

This community is found in the Shangi Derkie forest patch, which has a very steep slope. The altitudinal range of this community type lies between 2516 and 2947 ma.s.l. The community has six indicator species with significant indicator values, namely; *Erica arborea*, *Osyris quadripartita*, *Bersama abyssinica*, *Buddleja polystachya*, *Maesa lanceolata*, and *Rosa abyssinica*. The tree layer was dominated by *Ekebergia capensis* and *Lepidotrichilia volkensii*. The shrub layer was dominated by *Lippia adoensis*. The most dominant species in the herb layer include *Kniphofia foliosa*, *Kalanchoe petitiana*, *Verbascum sinaiticum* and *Trifolium decorum*. This community type is represented by 108 species, being the richest in a number of species among the five community types.

### *Discopodium penninervium-Echinops pappii* community type (C2)

The altitudinal range of this community type lies between 2375 and 2916 m a.s.l and located in Shangi Derkie, Aradie, Bunise, and Masikana Medhanyalm forest patches. The indicator

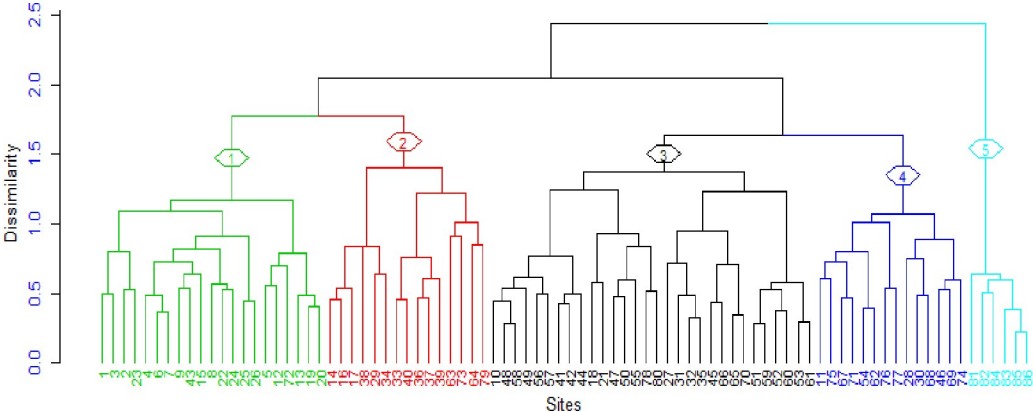

**Fig 3. Dendrogram of the vegetation data obtained from hierarchical cluster analysis of the study area** (1 = Community type 1, 2 = Community type 2, 3 = Community type 3, 4 = Community type 4 and 5 = Community type 5).

**Table 3. Indicator values of species in each community type (C1—C5) and their significance ($P < 0.05$).**

| Indicator Species | Indicator values (%) | | | | | |
|---|---|---|---|---|---|---|
| | **1** | **2** | **3** | **4** | **5** | **P value** |
| *Erica arborea* | **51.8** | 0 | 12.8 | 0 | 0 | 0.001 |
| *Osyris quadripartita* | **39** | 6.5 | 15.4 | 3.9 | 0 | 0.001 |
| *Bersama abyssinica* | 37.5 | 13.9 | 8.6 | 18.2 | 0 | 0.001 |
| *Buddleja polystachya* | 34 | 4.8 | 2.4 | 7.4 | 0 | 0.01 |
| *Rosa abyssinica* | 33.2 | 15.5 | 11.4 | 3.1 | 0 | 0.007 |
| *Maesa lanceolata* | 23.1 | 1.4 | 3.4 | 1.6 | 2.2 | 0.033 |
| *Discopodiumpenninervium* | 0.4 | **40.6** | 3.3 | 0 | 0 | 0.004 |
| *Echinops pappii* | 3 | **36.2** | 1.5 | 0.4 | 0 | 0.01 |
| *Laggera tomentosa* | 5 | 32.2 | 1.3 | 0 | 0 | 0.01 |
| *Olea europaea* | 2.2 | 4.4 | **47.2** | 0.5 | 0 | 0.001 |
| *Scolopia theifolia* | 0 | 0 | **43.7** | 11.1 | 0 | 0.001 |
| *Olinia rochetiana* | 19.7 | 1.5 | 42.1 | 15 | 0 | 0.001 |
| *Myrsine africana* | 23.2 | 0 | 28.7 | 2.5 | 0 | 0.028 |
| *Clutia abyssinica* | 9.9 | 17.9 | 28 | 0.2 | 0 | 0.046 |
| *Pittosporium viridiflorum* | 5.2 | 0.5 | 24.2 | 0.4 | 0 | 0.035 |
| *Euphorbia abyssinica* | 0 | 0.4 | 10.2 | **45.7** | 0 | 0.001 |
| *Prunus africana* | 3.8 | 0.3 | 13.5 | **30.5** | 0 | 0.019 |
| *Solanecio gigas* | 0.9 | 5.7 | 6.7 | 27.9 | 0 | 0.028 |
| *Embelia schimperi* | 1.1 | 0 | 0 | 27.8 | 0 | 0.005 |
| *Dodonaea anguistifolia* | 0.6 | 0.1 | 0.9 | 0 | **91.3** | 0.001 |
| *Acokanthera schimperi* | 0 | 0 | 0 | 0 | **83.3** | 0.001 |
| *Croton macrostachyus* | 1.5 | 2.4 | 0.4 | 0.2 | 56.7 | 0.001 |
| *Euclea schimperi* | 0 | 0 | 0 | 0 | 50 | 0.001 |
| *Acacia seyal* | 0 | 0 | 0 | 0 | 50 | 0.002 |
| *Combretum molle* | 0 | 0 | 0 | 0 | 50 | 0.001 |
| *Rhus vulgaris* | 0 | 0 | 0 | 0.7 | 45.2 | 0.001 |
| *Premna schimperi* | 0 | 0 | 0 | 0 | 33.3 | 0.008 |
| *Senna singueana* | 0 | 0 | 0 | 0 | 33.3 | 0.003 |

species are *Discopodium penninervium*, *Echinops pappii* and *Laggera tomentosa*. The most dominant shrub species were *Solanecio gigas*, *Maytenus arbutifolia*, *Solanum incanum*, *Rumex nervosus*, *Gnidia glauca* and *Acanthus sennii*. The tree layer was dominated by *Arundinaria alpina*. Liana such species such as *Asparagus africanus* and *Phytolacca dodecandra* were dominant. The herb layer was dominated by *Plectranthus assurgens* and *Carduus schimperi*. This community is comprised of 104 species.

### *Olea europaea* -*Scolopia theifolia* Community type (C3)

This community lies between altitudes 2475 and 2906 m above sea level. Similar to community 1 and 2, *Olea europaea* subsp. *cuspidata*—*Scolopia theifolia* Community type is also found in Bunise, shangi Derkie, Aradie, and Masikana Medihanyalm forest patches. The indicator tree and shrub species of this community are *Olea europaea* subsp. *cuspidata* and *Scolopia theifolia*, *Olinia rochetiana*, *Myrsine africana*, *Clutia abyssinica*, *Pittosporum viridiflorum*. The dominant species in the tree layer include *Apodytes dimidiata*. The shrub species such as *Rhus glutinosa*, subsp. *glutinosa*, *Dovyalis verrucosa*, and *Catha edulis* were highly dominant. The herb layer comprised of *Oplismenus hirtellus*, *Impatiens hochstetteri*, *Impatiens rothii* and *Justicia heterocarpa*. This community is comprised of 73 species.

**Table 4. Diversity and evenness of plant communities.**

| Plant community types | Richness | Shannon_Evenness (J) | Shannon Diversity (H) |
|---|---|---|---|
| Comminty 1 | 108 | 0.837764 | 3.922522 |
| Comminty 2 | 104 | 0.83718 | 3.888192 |
| Comminty 3 | 73 | 0.863749 | 3.705881 |
| Comminty 4 | 84 | 0.878393 | 3.891999 |
| Comminty 5 | 31 | 0.840046 | 2.884706 |

### *Euphorbia abyssinica-Prunus africana* community type (C4)

This community type is located between altitudes 2455 and 2733 m asl, and which is found in 3 forest patches: Bunise, Shangi Derkie, Aradie, and Masikana. The indicator species are *Euphorbia abyssinica* and *Prunus africana*. The shrub layer includes *Solanecio gigas* and liana species *Embelia schimperi* were also dominant. The herb layer was dominated by *Hypoestes for-skaolii*. It consists of 84 species.

### *Dodonaea anguistifolia-Acokanthera schimperi* community type (C5)

Community 5 found in the Samana forest patch. This community is established at lower elevations (1881–2110 m a.s.l.). *Dodonaea anguistifolia* and *Acokanthera schimperi* are the characteristic species of the shrub, and tree layers, respectively. Other dominant tree species include *Croton macrostachyus*, *Acacia seyal*, and *Combretum molle*. The shrub layer was dominated by *Euclea schimperi*. Dominant species of the herb layer were *Bidens macroptera* and *Satureja abyssinica*.

### Diversity and evenness of plant communities

Shannon's diversity indices showed that community 1 had the highest species diversity and richness followed by community 2, while community 5 had the lowest species diversity and richness. On the other hand, community 4 had the highest evenness value followed by communities 3 and 5. Community 1 and 2 had the least evenness value (Table 4).

### Similarities between plant communities

Results of plant community similarity indicated that communities 2 and 3, communities 1 and 2, communities 1 and 3 have the same similarity index values. The least similarity was between community 1 and 5, community 3 and 5, community 4 and 5 (Table 5).

**Table 5. Similarity indices of plant communities (C1—C5) in the study area.**

| | Plant community types | | | | |
|---|---|---|---|---|---|
| | **C1** | **C2** | **C3** | **C4** | **C5** |
| C1 | - | | | | |
| C2 | 0.72 | - | | | |
| C3 | 0.72 | 0.74 | - | | |
| C4 | 0.60 | 0.64 | 0.68 | - | |
| C5 | 0.20 | 0.25 | 0.20 | 0.20 | - |

Note: C1: *Erica arborea-Osyris quadripartita*, C2: *Discopodium penninervium-Echinops pappii*, C3: *Olea europaeasubsp. cuspidata—Scolopia theifolia*, C4: *Euphorbia abyssinica—Prunus africana*, C5: *Dodonaea anguistifolia—Acokanthera schimperi* plant community types.

**Table 6. Pearson correlation coefficients between environmental variables in Dega Damot District, Northwestern Ethiopia.**

| - | Slo | Elev | MOc | BD | Clay | PH | EC | OM | CEC | N | P |
|---|---|---|---|---|---|---|---|---|---|---|---|
| Slo | 1 | | | | | | | | | | |
| Elev | -0.08 | 1 | | | | | | | | | |
| MOc | 0.18 | 0.26* | 1 | | | | | | | | |
| BD | 0.01 | -0.22* | -0.22* | 1 | | | | | | | |
| Clay | 0.264* | 0.35** | 0.48** | -0.37** | 1 | | | | | | |
| pH | 0.28* | -0.4** | 0.03 | 0.11 | -0.96* | 1 | | | | | |
| EC | 0.04 | -0.06 | 0.17 | -0.15 | 0.23* | 0.09 | 1 | | | | |
| OM | 0.26 | 0.18** | 0.67** | -0.3** | 0.71** | 0.14 | 0.28* | 1 | | | |
| CEC | 0.01 | 0.01 | 0.56** | -0.21* | 0.27* | 0.191 | 0.17 | 0.58** | 1 | | |
| N | 0.06 | 0.40** | 0.52** | -0.25* | 0.40** | -0.01 | 0.08 | 0.65** | 0.51** | 1 | |
| P | -0.19 | -0.06 | 0.38** | -0.04 | 0.19** | 0.38** | 0.15 | 0.44** | 0.47** | 0.25* | 1 |

*. Correlation is significance at the 0.05 level (2-tailed)

**. Correlation is significant at the 0.01 level (2-tailed), Slo, slope; Elev, elevation; Moc, Moisture content; BD, Bulk density.

## Correlation of environmental variables

Pearson's correlation matrix of the environmental factors is shown in Table 6. Altitude was positively correlated with moisture content, nitrogen and sand are negatively correlated with pH and bulk density. Nitrogen and phosphorus are positively correlated with soil moisture content and organic matter. CEC is positively correlated with organic matter, nitrogen and phosphorus. Soil pH shows a positive correlation with phosphorus while EC shows a positive correlation with organic matter (see Table 6).

## Relationship between plant community types and environment variables

**RDA ordination.** Out of the 10 environmental variables included in ordination analysis (slope, bulk density, moisture content, elevation, clay, pH, P, EC, OM and CEC), only elevation, moisture content, OM, slope, pH, EC, clay and phosphorus had significant ($p < 0.05$) correlated on species compositions of communities and their distributions as shown in Table 7 and Fig 4. We described several of these variables are significant but none of them account for much variability except elevation. A high sum of squeres would indicate a lot of variability in the data, while a low sum of squeres (most of the measurement close to mean) would indicate a low amount of variability (Table 7). The RDA diagram indicated that the first axis was primarily correlated with elevation, clay, and moisture content, while the second axis was correlated with P, OM and moisture content (Fig 4, Table 8). The eigenvalue for the first and second axis were 0.54 and 0.19, respectively. The first axis eigenvalue which closer to 1 and show distribution of the plant communities along the axis is good. The cumulative proportion variance explained by the first six RDA axis of the joint biplot was 92%. The variation in patterns of plant species distribution and plant community formation for the first and second axis were 46%, and 16%, respectively. This showed that 62% of the variation in patterns of plant species distribution and plant community formation was explained by axis one and two.

Consequently, the first two axes are sufficient to reflect the relationship between species and environmental factors. In the first RDA axis, elevation was the most important variable and separated communities at high elevation (communities 1, 2, 3, and 4) from communities distributed at lower elevation (communities 5) on the ordination axis (Fig 4). On the other hand, in the second axis, communities 2 and 3 were located on sites with higher organic matter

**Table 7. Result of function donis test of environmental variables (significant environmental variables are indicated by Asterix at their p-value) the study area.**

| Environmental variables | Df | Sums of Sqs. | Mean Sqs | F. Model | R$^2$ | Pr(>F) |
|---|---|---|---|---|---|---|
| Slope (%) | 1 | 0.5315 | 0.53153 | 2.4132 | 0.02232 | 0.007 ** |
| Altitude(m asl) | 1 | 2.6735 | 2.67347 | 12.1376 | 0.11225 | 0.001*** |
| MOC (%) | 1 | 0.9862 | 0.98618 | 4.4773 | 0.04141 | 0.001 *** |
| Bulk Density | 1 | 0.3151 | 0.31505 | 1.4303 | 0.01323 | 0.137 |
| Clay | 1 | 0.3822 | 0.38217 | 1.7351 | 0.01605 | 0.042* |
| pH | 1 | 0.5543 | 0.55432 | 2.547 | 0.02327 | 0.002 *** |
| EC | 1 | 0.6313 | 0.63132 | 2.7876 | 0.02651 | 0.002 ** |
| OM | 1 | 0.4122 | 0.51573 | 1.8716 | 0.01731 | 0.028* |
| CEC | 1 | 0.3469 | 0.2462 | 1.5748 | 0.01034 | 0.086 |
| P | 1 | 0.4745 | 0.4745 | 2.1544 | 0.0199 | 0.010 * |
| Residuals | 75 | 16.5197 | 0.22026 | | 0.69361 | |
| Total | 85 | 23.8170 | | | 1.00000 | |

Signif. codes: 0

'***' 0.001

'**' 0.01

'*' 0.05 '.' 0.1 '' 1

and moisture content. The values of the different environmental variables average for each plant community type are presented in Table 9.

## Discussion

We found relatively high plant species richness in Dega Damot district forest patches. However, the study area is fewer species-rich, especially in woody species than similar studies conducted in the Afromontane forest in Ethiopia, e.g. Tara Gedam and Abebaye forests [45], Afromontane forest patches of Awi zone [46] It has higher species richness than other dry Afromontane forests in Ethiopia such as the Zengena forest [47], Kuandisha forest [48], and Amoro forest [26]. Besides, the number of woody species recorded in this study is more or less similar to that of the Zege Peninsula forest [49] whereas [50] recorded low woody species in the vegetation of Kalfou forest reserve, Cameroon. The difference in species composition among the different areas is due to the number of plots sampled and its size can somehow explain this heterogeneity of the species richness. [51] reported that forests with a high degree of human interference and disturbances show relatively lower species richness than others. Thus, the present study suggests that the Dega Damot district had relatively high species composition, compared to other similar vegetation types in Ethiopia.

The dominance of Asteraceae was reported from other studies in dry evergreen Afromontane forests vegetation type [23, 52, 53]. [54] also reported that Asteraceae was the richest family in terms of species numbers for the flora of Ethiopia and Eretria. The dominance of Asteraceae in the present study may indicate that the forest patches might have been under a certain level of disturbances. As reported by [55] and [54], Asteraceae usually have a preference for open and disturbed lands to grow.

According to [56] the proportion of endemic plant species in the montane forests of Ethiopia is high, between 11–15% of the total number of species. Similarly, the results from the study area showed that relatively high endemicity (10.6%). Ethiopia is one of the centers of plant endemism in East Africa [57]. This study is in line with the typical feature of Afromontane forests that house numerous endemic species [58]. However, the Ethiopian Afromontane

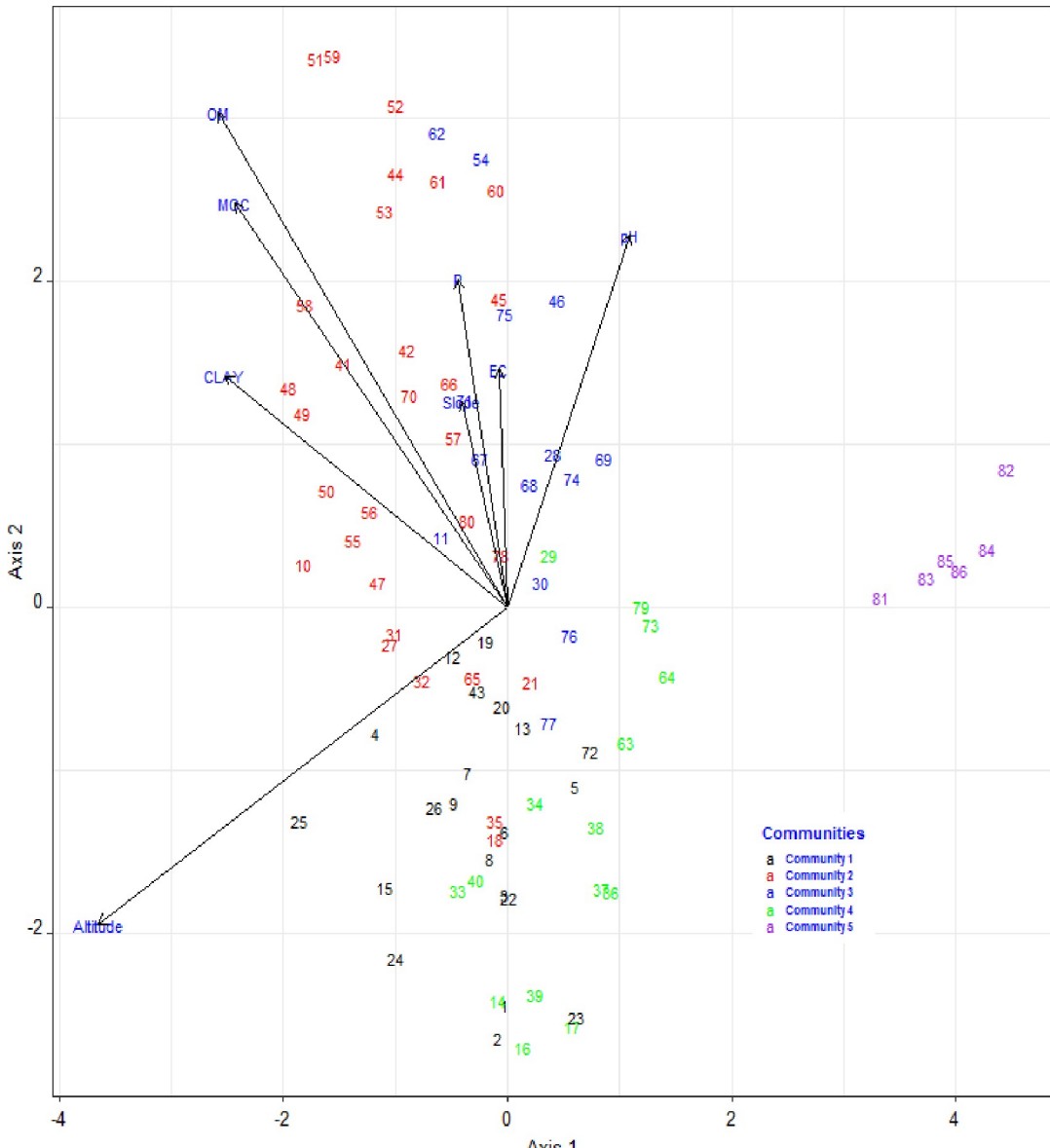

**Fig 4. Redundancy analysis (RDA) ordination graph significant environmental variables (p < 0.05) and the plant community in the study area.** The arrows in the diagram stand for the environmental factors, the length of each arrow indicates the contribution of the factor to ordination axes, The numbers refer to quadrat number, and the angle between the arrows and he axes indicates the correlation between the variable and the ordination axe.

forests are one of the most degraded forests and continuously shrinking mainly due to anthropogenic disturbances [59, 60].

## Plant communities

Based on the hierarchical cluster analysis result, 5 plant communities were identified in the study area. However, the location of the 3 community types (community 2, 3, and 4) had overlapping ranges of elevation. Elevation represents a complex gradient combination of many different environmental factors such as topography, soil, moisture and climate [59] So that it is difficult to separate other environmental factors [61].

**Table 8. Biplot scores for constraining variables.**

| Variables | RDA1 | RDA2 | RDA3 | RDA4 | RDA5 | RDA6 |
|---|---|---|---|---|---|---|
| Slope | 0.0025 | 0.13 | -0.5861 | 0.496 | -0.4817 | 0.045 |
| Altitude | -0.9492 | -0.29 | 0.0079 | 0.082 | 0.044 | -0.049 |
| Moisture content | -0.3664 | 0.67 | -0.217 | 0.072 | 0.092 | 0.172 |
| Clay | -0.4319 | 0.53 | -0.2948 | 0.084 | 0.0086 | -0.633 |
| PH | 0.2330 | 0.30 | -0.3008 | -0.594 | -0.52 | 0.283 |
| EC | -0.0504 | 0.42 | -0.2941 | -0.098 | 0.654 | 0.206 |
| OM | -0.3424 | 0.86 | -0.2748 | 0.144 | - 0.008 | -0.068 |
| P | -0.1120 | 0.74 | 0.4181 | -0.322 | -0.16 | 0.217 |
| Eigenvalue | 0.5392 | 0.1887 | 0.1238 | 0.0933 | 0.062 | 0.0634 |
| Proportion Explained | 0.4612 | 0.1614 | 0.1059 | 0.077 | 0.05829 | 0.053 |
| Cumulative Proportion | 0.4612 | 0.6226 | 0.7285 | 0.80 | 0.920 | 0.920 |

The vegetation of the area is classified as dry evergreen Afromontane forest and grassland complex (DAF) [28] and dominated by the species *Olea europaea* subsp. *cuspidata*, *Juniperus procera*, *Discopodium penninervium*, *Pittosporum viridiflorum*, *Erica arborea*, *Hypericum revolutum*, *Dodonaea angustifolia*, *Acacia abyssinica*, *Bersama abyssinica*, and *Prunus africana*. The *Olea europaea* subsp. *cuspidata*, indicator species was also mentioned to form the upper canopies of different dry Afromontane forests in Ethiopia [19, 21].

The identified plant communities are characterized by their different floristic composition. This could be attributed to variations in environmental factors. According to [62, 63], the variation species composition among plant communities probably associated with the effects of environmental factors. Nevertheless, species diversity and richness were not the same among plant communities. For instance, the highest species richness and diversity were recorded in community 1. On the contrary, community 5 had the least species richness and diversity than the remaining community types. The reason for high species diversity and richness of community 1 may be due to the highest altitudinal ranges in which this community is found (2516–2947 m). Also, community 5 type is the most disturbed community due to settlement, agricultural expansion, overgrazing by livestock and crossing road.

Anthropogenic land-use change is one of the most important factors contributing to global change [64], and changes in landscape structure and agricultural land-use intensity are likely to influence plant community composition and species richness. Changes in land-use practice

**Table 9. Post-hoc comparison of means between environmental variables and plant communities.**

| | Plant community types | | | | |
|---|---|---|---|---|---|
| Environmental variable | C1 | C2 | C3 | C4 | C5 |
| Elevation | 2762.67[a] | 2783.60[a] | 2727.17[a] | 2593.64[b] | 2021.33[c] |
| Moisture content | 10.363[b] | 10.710b[a] | 14.782[a] | 13.908b[a] | 4.697[c] |
| Clay | 21.143[bc] | 27.533[ab] | 20.867[bc] | 19.000[c] | 33.000[a] |
| PH | 6.3419[b] | 6.39733[ab] | 6.44500[ab] | 6.60357[a] | 6.63000[a] |
| Organic matter | 7.561[ab] | 6.551[b] | 10.244[a] | 10.332[a] | 3.485[c] |
| CEC | 63.111[ab] | 58.069[cb] | 66.780[ab] | 72.433[a] | 51.088[c] |
| N | 0.38205[ab] | 0.32040[b] | 0.42547[a] | 0.41150[a] | 0.15967[c] |
| P | 9.048[b] | 18.000[ab] | 17.367[ab] | 24.071[a] | 6.167[b] |

Note: Values in a row with different letters are significantly different (P<0.05).

may result in continuously or more abruptly deteriorating environmental conditions for some plant species, causing their decline in abundance and distribution [65]. For instance, over the last 31 years, the forest cover of Dega Damot district has decreased while areas under farmland, grazing, and settlements have increased [25]. Therefore, these anthropogenic land-use changes might influence the plant community formation of the present study in a different plant species richness and diversity. The result of the pairwise comparison of Sorensen's similarity coefficient index in species composition between the 3 plant communities (Community 1, 2 and 3) showed high similarity. These due to the location of this community have relatively similar environmental factors (soil and altitudinal range). Community 5 located at the lowest elevation with low content of organic matter, soil moisture content and which may have a less floristic similarity from the remaining plant community types (Community 1, 2 and 3).

## Environmental factors and plant community relationship

The distribution of plant communities in the study area reflects the combined influence of altitude and soil factors. Based on the results of RDA, elevation was the major environmental variable in explaining variations in plant species distribution and patterns of plant community formation although there are overlaps among some community types. This might be associated with a continuous change in environmental variables along the elevational gradient [63, 66]. Other studies conducted in Ethiopia [see 18, 23, 67] also noted that altitude is the most important environmental variable for the determinant of vegetation variation. Moreover [68], reported that vegetation distribution influenced by elevation. Besides, elevation is a vital environmental factor that affects the atmospheric pressure, moisture, and temperature which have a strong influence on the growth and development of plants and the distribution of vegetation [69].

Moreover, organic matter was also the most important constraining variable in plant community formation of the study area. Research in the Jibat forest, Ethiopia, by [19] reported that plant distributions were affected by the organic matter at a higher elevation. SOM, its role in soil structure and moisture retention capability are well known [70, 71]; these effects may account for its relevance for determining plant species distribution and community formation in the study area, besides its role as a source of soil nutrients to plants. The mineralization of organic matter is a contributing factor in supplying available mineral nutrients for plant use and through decomposition process the available nitrogen.

Similar to the present findings, nitrogen and P were also attributed to have significant effects on species compositions of plant communities in remnant Afromontane forests on the central plateau of Shewa [20]. Phosphorus is an essential nutrient for the growth of plant. This result is also in agreement with past similar works in in Afromontane and transitional rainforest vegetation of southwestern Ethiopia [67]. Furthermore, a study conducted in Brazil by [72], P was the main factor that distinct plant communities.

Topographic features associated with soil properties are strongly correlated with species distribution and plant community structure on a local scale [33]. For instance, the *Dodonaea anguistifolia-Acokanthera schimperi* community, with a low content of organic matter, soil moisture content, and N, is differentiated from the other community types in the study area. The anthropogenic disturbances are higher at the lower altitudes (selective cutting of trees, grazing by livestock and expanding of farmlands) and the temperature is also higher at these altitudes. Generally, in the study area, the soil organic matter is higher at higher altitudes. This might be due to the decreased decomposition of organic matter and the long-term accumulation of organic matter [73]. Much of the soil nitrogen is obtained from organic matter [74] and hence is bound in higher quantities at higher elevations.

## Conclusion

Five plant communities were identified from this study. Among all of the investigated environmental factors, elevation, moisture content, slope, pH, EC, clay, OM, and phosphorus were found to significantly explain variation in species composition and community formation in the study area. Elevation was the most important environmental factor influencing species distribution and community formation. The lower elevation resulted in a decline in species richness, diversity, organic matter, moisture content, especially in the community 5 in the forest. The anthropogenic disturbances present in this community type may also contribute to the low species richness and diversity recorded. Also, *Dodonaea anguistifolia-Acokanthera schimperi* community type is not correlated to all the measured environmental variables. Therefore, it can be concluded that other environmental factors may influence the plant community formation, which is required to be additionally studied.

## Supporting information

**S1 Appendix. Floristic list of Dega Damot district.**
(DOCX)

**S2 Appendix. Descriptive statistical analysis of the topography and soil data.**
(DOCX)

## Acknowledgments

The authors express their deepest thanks to Agriculture and Rural Development Office of the Dega Damot District as well as to the respective chairpersons for their assistance for field data collection. We extend our sincere thanks to the anonymous reviewers of PLOS ONE for their comments.

## Author Contributions

**Conceptualization:** Liyew Birhanu, Tamrat Bekele, Binyam Tesfaw, Sebsebe Demissew.

**Data curation:** Liyew Birhanu.

**Formal analysis:** Liyew Birhanu.

**Investigation:** Liyew Birhanu.

**Methodology:** Liyew Birhanu.

**Software:** Liyew Birhanu.

**Supervision:** Tamrat Bekele, Binyam Tesfaw, Sebsebe Demissew.

**Validation:** Tamrat Bekele, Binyam Tesfaw, Sebsebe Demissew.

**Visualization:** Liyew Birhanu.

**Writing – original draft:** Liyew Birhanu.

**Writing – review & editing:** Liyew Birhanu, Tamrat Bekele, Binyam Tesfaw, Sebsebe Demissew.

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
