## [Decision Letter · Decision Letter 0]

11 Dec 2020

PONE-D-20-31398

Relationships between topographic factors, soil and plant communities in a Dry Afromontane forest patches of Northwestern Ethiopia

PLOS ONE

Dear Dr. Birhanu,

Thank you for submitting your manuscript to PLOS ONE. After careful consideration, we feel that it has merit but does not fully meet PLOS ONE’s publication criteria as it currently stands. Therefore, we invite you to submit a revised version of the manuscript that addresses the points raised during the review process.

We look forward to receiving your revised manuscript.

Kind regards,

Mehdi Heydari

Academic Editor

PLOS ONE

Journal Requirements:

2. PLOS specifies that experiments, statistics, and other analyses are performed to a high technical standard; sample sizes are large enough to produce robust results; and methods are described in sufficient detail to allow another researcher to reproduce the experiment (http://journals.plos.org/plosone/s/criteria-for-publication#loc-3). As such, we ask you to amend your Methods section to provide detail on the instrumentation and methods of analysis used. Mnaufacturer information should be provided for instruments and chemicals used in the study.

3. In your Methods section, please provide additional information regarding the permits you obtained to collect samples for the present study. Please ensure you have included the full name of the authority that approved the field site access and, if no permits were required, a brief statement explaining why.

"The authors express their deepest thanks to Addis Ababa University for financial support. Agriculture and Rural Development Office of the Dega Damot District as well as to the respective chairpersons is also highly acknowledged."

"No source of fund"

6. Please amend your list of authors on the manuscript to ensure that each author is linked to an affiliation. Authors’ affiliations should reflect the institution where the work was done (if authors moved subsequently, you can also list the new affiliation stating “current affiliation:….” as necessary).

7. Please amend either the abstract on the online submission form (via Edit Submission) or the abstract in the manuscript so that they are identical.

8. We note that Figures 1 and 2 in your submission contain map images which may be copyrighted. All PLOS content is published under the Creative Commons Attribution License (CC BY 4.0), which means that the manuscript, images, and Supporting Information files will be freely available online, and any third party is permitted to access, download, copy, distribute, and use these materials in any way, even commercially, with proper attribution. For these reasons, we cannot publish previously copyrighted maps or satellite images created using proprietary data, such as Google software (Google Maps, Street View, and Earth). For more information, see our copyright guidelines: http://journals.plos.org/plosone/s/licenses-and-copyright.

(1) You may seek permission from the original copyright holder of Figures 1 and 2 to publish the content specifically under the CC BY 4.0 license. 

9. Please include your tables as part of your main manuscript and remove the individual files. Please note that supplementary tables (should remain/ be uploaded) as separate "supporting information" files.

10. Please include captions for your Supporting Information files at the end of your manuscript, and update any in-text citations to match accordingly. Please see our Supporting Information guidelines for more information: http://journals.plos.org/plosone/s/supporting-information.

11. We note in your Data Availability statement you have advised "No - some restrictions will apply"

Could you please  clarify the nature of these restrictions, ie. If due to ethical or legal reasons.

PLOS defines a study's minimal data set as the underlying data used to reach the conclusions drawn in the manuscript and any additional data required to replicate the reported study findings in their entirety. All PLOS journals require that the minimal data set be made fully available. For more information about our data policy, please see http://journals.plos.org/plosone/s/data-availability.

12. We noticed you have some minor occurrence of overlapping text with the following previous publication(s), which needs to be addressed:

https://link.springer.com/article/10.1007/s11676-010-0089-9

https://www.ajol.info//index.php/sinet/article/view/18247

http://wiredspace.wits.ac.za/handle/10539/9163

https://www.tandfonline.com/doi/full/10.1080/20964129.2017.1385004

In your revision ensure you cite all your sources (including your own works), and quote or rephrase any duplicated text outside the methods section. Further consideration is dependent on these concerns being addressed.

Reviewers' comments:

Reviewer's Responses to Questions

**Comments to the Author**

1. Is the manuscript technically sound, and do the data support the conclusions?

Reviewer #1: Yes

Reviewer #2: Yes

2. Has the statistical analysis been performed appropriately and rigorously? 

Reviewer #1: Yes

Reviewer #2: Yes

3. Have the authors made all data underlying the findings in their manuscript fully available?

Reviewer #1: No

Reviewer #2: No

4. Is the manuscript presented in an intelligible fashion and written in standard English?

Reviewer #1: Yes

Reviewer #2: Yes

5. Review Comments to the Author

Reviewer #1: PONE-D-20-31398

It is rather interesting and refreshing to read a paper about the characterization of plant communities. This is of obvious importance in this region of Ethiopia, where plant communities have not been previously categorized.

The point of this study was to document distinct plant communities in the Dega Damot District, and to relate these to various environmental factors.

One of the difficulties in establishing vegetation types (communities) is natural variability. Thus, it is important to sample from a large number of habitats in order to determine how reproducible a particular community composition is, how much variability occurs within a single community type, and how much control various environmental factors exert on community structure.

Based on the description of the sampling effort, it is difficult for me to determine how much sampling actually occurred within a community type. The authors refer to “forest patches”, but just what these are is not clear. “Sampling sites were arranged along transects from the forest patches”. There were 6 transects in both in Shangi Derke and Bunsie patches. There were 4 transects in the Aradie patch. And there were 3 transects in both Masikana and Samana patches. Were these patches already known to contain distinct plant communities so the level of replication within a community type could be controlled? If not, how did the authors control the level of replication within a community type?

By my reading, it appears that 86 plots were sampled in each of the 5 patches, for a total of 430 plots. Is that true?

The authors wrote “The first transect and plot were selected purposely which are free from any anthropogenic disturbances.” How was that determined? Does that mean that some plots were disturbed? If so, wouldn’t disturbance influence the structure of the plant community, and wouldn’t this bias our understanding of these communities?

There were apparently 5 smaller plots per larger plot for herbaceous plant sampling for a total of 2,150 small plots. Is that true? Also, please specify when the smaller plots were sampled. Is it possible for sampling time to affect the herbaceous species that were observable? Please specify.

Because the spatial and temporal aspects of the vegetation and soil sampling schemes were not clear to me, perhaps a figure illustrating them would be helpful.

I am not sure how the cluster analysis was performed. The authors wrote that clustering was performed based on attributes and floristic similarities. What is meant by “attributes”? Does that include physical traits of the habitat? Please be more specific about the exact variables that were used in the cluster analysis.

Some of the information given in the results is repeated in the discussion unnecessarily. To eliminate this repetition, it may be better to have a single section of text called Results and Discussion rather than separate sections.

It would be nice if the location of the forest patches could be placed on Figure 2.

Table 7 indicates that several of these variables are significant, but the authors should be cautious as none of them account for much variability (sums of squares are all low). This should be indicated in the text.

Reviewer #2: This work by Birhanu et al. describes the results of comprehensive vegetation surveys across 86 sites in Northwestern Ethiopia. They evaluated how climatic and topoedaphic variation determine patterns of plant community composition and species abundance across the region. They highlighted clusters of community types based on hierarchical analysis and a series of standard community composition metrics. These species-rich sites were compared to other high diversity forest areas in the region, as well as to forests in Cameroon. The methods used in this study were appropriate and rigorously applied, and the conclusions they drew in their analysis are supported by their results. This well-written work appropriately characterizes the roles of soils, climate and topography in driving community composition across the region, and I recommend it be accepted for publication.

6. PLOS authors have the option to publish the peer review history of their article (what does this mean?). If published, this will include your full peer review and any attached files.

Reviewer #1: No

Reviewer #2: No

---

## [Author Response · Author response to Decision Letter 0]

2 Feb 2021

We would like to thank the academic editor and reviewers for their efforts in reviewing the manuscript and providing valuable suggestions for improving the article. All comments and suggestions were addressed

---

## [Editor Report · Decision Letter 1]

17 Feb 2021

Relationships between topographic factors, soil and plant communities in a Dry Afromontane forest patches of Northwestern Ethiopia

PONE-D-20-31398R1

Dear Dr. Birhanu,

We’re pleased to inform you that your manuscript has been judged scientifically suitable for publication and will be formally accepted for publication once it meets all outstanding technical requirements.

Kind regards,

Mehdi Heydari

Academic Editor

PLOS ONE

Additional Editor Comments (optional):

Dear author

I am pleased to inform you that your manuscript has been accepted for publication.  

We appreciate you submitting your manuscript to PLOS ONE and hope you will consider us again for future submissions.

Regards

MH

ACADEMIC EDITOR
---

## [Editor Report · Acceptance letter]

22 Feb 2021

PONE-D-20-31398R1 

Relationships between topographic factors, soil and plant communities in  a Dry Afromontane forest patches of  Northwestern Ethiopia 

Dear Dr. Birhanu:

I'm pleased to inform you that your manuscript has been deemed suitable for publication in PLOS ONE. Congratulations! Your manuscript is now with our production department. 

Kind regards, 

on behalf of

Dr. Mehdi Heydari 

Academic Editor

PLOS ONE